# AI and Machine Learning in Biology: From Genes to Proteins

**DOI:** 10.3390/biology14101453

**Published:** 2025-10-20

**Authors:** Zaw Myo Hein, Dhanyashri Guruparan, Blaire Okunsai, Che Mohd Nasril Che Mohd Nassir, Muhammad Danial Che Ramli, Suresh Kumar

**Affiliations:** 1Department of Basic Medical Sciences, College of Medicine, Ajman University, Ajman P.O. BOX 346, United Arab Emirates; z.hein@ajman.ac.ae; 2Center of Medical and Bio-Allied Health Sciences Research (CMBHSR), Ajman University, Ajman P.O. BOX 346, United Arab Emirates; 3Department of Diagnostic and Allied Health Science, Faculty of Health and Life Sciences, Management and Science University, Seksyen 13, Shah Alam 40100, Malaysia; dhanya9874@gmail.com (D.G.); hobi66blaire@gmail.com (B.O.); 4Department of Anatomy and Physiology, School of Basic Medical Sciences, Faculty of Medicine, University Sultan Zainal Abidin, Kuala Terengganu 20400, Malaysia; nasrilnassir@unisza.edu.my

**Keywords:** artificial intelligence, genomics, protein structure prediction, deep learning, precision medicine, convolutional neural networks, multi-omics integration

## Abstract

**Simple Summary:**

Artificial intelligence (AI) and machine learning (ML), particularly deep learning methods like neural networks and transformers, are revolutionizing biology by analyzing massive amounts genetic and protein data. These technologies enhance the prediction of gene functions, the identification of disease-causing mutations, and accurate protein structure modeling, showcased by breakthroughs such as AlphaFold for protein folding and DeepBind for DNA regulatory element detection. By linking genomic data with protein functions, AI accelerates the journey from DNA sequences to functional molecules, aiding drug discovery and personalized medicine. Key advances impact epigenetics, single-cell disease analysis, and novel protein and drug design. However, challenges remain, including the need for large, high-quality data, model interpretability, and ethical issues like privacy and bias. Future progress relies on integrating complex biological data, improving transparency, ensuring fairness, and ethical training, potentially transforming healthcare with personalized, responsible AI-driven solutions.

**Abstract:**

Artificial intelligence (AI) and machine learning (ML), especially deep learning, have profoundly transformed biology by enabling precise interpretation of complex genomic and proteomic data. This review presents a comprehensive overview of cutting-edge AI methodologies spanning from foundational neural networks to advanced transformer architectures and large language models (LLMs). These tools have revolutionized our ability to predict gene function, identify genetic variants, and accurately determine protein structures and interactions, exemplified by landmark milestones such as AlphaFold and DeepBind. We elaborate on the synergistic integration of genomics and protein structure prediction through AI, highlighting recent breakthroughs in generative models capable of designing novel proteins and genomic sequences at unprecedented scale and accuracy. Furthermore, the fusion of multi-omics data using graph neural networks and hybrid AI frameworks has provided nuanced insights into cellular heterogeneity and disease mechanisms, propelling personalized medicine and drug discovery. This review also discusses ongoing challenges including data quality, model interpretability, ethical concerns, and computational demands. By synthesizing current progress and emerging frontiers, we provide insights to guide researchers in harnessing AI’s transformative power across the biological spectrum from genes to functional proteins.

## 1. Introduction

### 1.1. Genomics and Protein Structure Prediction: A Unified Frontier Enabled by Deep Learning

Computational biology combines advanced computing with biological research to explore complex living systems, particularly in genomics and protein structure prediction [1]. Within this interdisciplinary realm, genomics and protein structure prediction represent two pivotal, yet intrinsically linked, areas of research. The journey from genetic information encoded in DNA to the functional machinery of proteins is a central dogma of molecular biology: DNA is transcribed into RNA, which is then translated into protein sequences. The linear sequence of amino acids in a protein subsequently folds into a unique 3D structure, which in turn determines its biological activity. Understanding this intricate flow of information from gene to protein structure to function is paramount for advancing our knowledge of biological systems and developing novel therapeutic interventions. This review combines genomics and protein structure prediction into a single, cohesive narrative due to their inherent biological interconnectedness and the synergistic role AI and ML plays in bridging these domains. The rationale is rooted in the central dogma of molecular biology: genomic information (DNA/RNA sequences) directly encodes the amino acid sequences of proteins, and these sequences, in turn, determine the protein’s three-dimensional structure, which is crucial for its function. AI and ML provide the computational framework to traverse this biological pathway, enabling a holistic understanding of biological systems from the genetic blueprint to the functional molecular machinery. By treating these fields jointly, we can better illustrate how advancements in one area, driven by deep learning, often directly impact and accelerate progress in the other, leading to a more comprehensive and integrated view of biological processes and disease mechanisms. Since the 1990s, machine learning has evolved from basic neural networks analyzing gene expression data to sophisticated deep learning algorithms. Models such as convolutional neural networks (CNNs), recurrent neural networks (RNNs), and transformers now detect complex patterns in genomic and proteomic datasets, enabling accurate predictions [2].

In 2015, researchers from Harvard and MIT developed DeepBind, a groundbreaking deep learning algorithm that identifies RNA-binding protein sites, revealing previously unknown regulatory elements in the genome [3]. Scientists increasingly rely on such algorithms to address biological challenges, from predicting protein structures to identifying disease-causing mutations. For example, DeepMind’s AlphaFold uses advanced neural networks to accurately predict proteins’ three-dimensional structures, opening new frontiers in structural biology [4]. These advancements have driven significant progress in genomics, medical diagnosis, and drug discovery. The use of AI and ML in computational biology has resulted in noteworthy breakthroughs spanning diverse niches, like genomics, medical diagnosis, and drug discovery. AI enables precise analysis of genomic data, identifying disease-causing mutations and supporting the development of personalized treatments. It also predicts functional pathways for new drugs, streamlining target identification and reducing reliance on trial-and-error experiments. By analyzing vast genomic, proteomic, and other biological datasets, deep learning uncovers subtle patterns often missed by traditional statistical methods, enhancing our understanding of biological systems.

The growing demand for personalized medicine and efficient drug discovery drives the adoption of AI in life sciences [5]. However, challenges remain. AI algorithms require large, high-quality datasets, which can be scarce in some biological fields [3]. Additionally, interpreting their results is complex, as they detect subtle patterns that may not align with traditional biological models.

Despite these challenges, AI has the potential to transform computational biology by deepening our understanding of biological systems and improving healthcare outcomes. This review explores its applications, addresses associated challenges, and highlights key advancements, such as AlphaFold and DeepBind, and their potential impact on personalized medicine and drug discovery in the coming years.

### 1.2. Brief History and Evolution of Deep Learning

The journey of deep learning, from its theoretical origins to its current state as a transformative technology, is marked by periods of intense research and significant breakthroughs. Rina Dechter introduced the term “deep learning” to the machine learning community in 1986, and Igor Aizenberg and colleagues applied it to artificial neural networks in 2000, focusing on Boolean threshold neurons [5]. The concept originated in 1943, when Warren McCulloch and Walter Pitts developed a computer model based on human neural networks, using “threshold logic” to simulate cognitive processes [6]. Since then, deep learning has evolved continuously, with brief setbacks during the “AI Winters” (periods of reduced funding and interest in AI research) [5]. Table 1 outlines the history and evolution of deep learning. In 1943, Warren McCulloch and Walter Pitts pioneered neural networks with a computational model called threshold logic, using mathematical algorithms to mimic cognitive processes [6]. In 1958, Frank Rosenblatt developed the perceptron, a two-layer neural network for pattern recognition based on simple arithmetic operations. He also proposed adding more layers, though practical implementation was delayed until 1975.

In 1980, Kunihiko Fukushima introduced the Neocognitron, a hierarchical, multilayered neural network that excelled in handwriting and pattern recognition tasks. By 1989, researchers developed algorithms for deep neural networks, though their lengthy training times (often days) limited practicality. In 1992, Juyang Weng’s Cresceptron enabled automated 3D object recognition in complex scenes, advancing neural network applications.

In the mid-2000s, Geoffrey Hinton and Ruslan Salakhutdinov’s seminal paper popularized deep learning by demonstrating the effectiveness of layer-by-layer neural network training [5]. In 2009, the NIPS Workshop on Deep Learning for Speech Recognition showed that pre-training could be skipped with large datasets, significantly reducing error rates. By 2012, deep learning algorithms achieved human-level performance in pattern recognition tasks, marking a major milestone in the field.

In 2014, Google acquired DeepMind, a UK-based AI startup, for £400 million, accelerating AI research advancements. In 2015, Facebook implemented DeepFace, a deep learning system with 120 million parameters, enabling accurate automatic tagging and identification in photographs. In 2016, DeepMind’s AlphaGo defeated professional Go player Lee Sedol in a highly publicized Seoul tournament, showcasing deep learning’s capabilities. By 2024, transformer-based models like AlphaFold3 predicted protein complexes and ligand interactions, while genomic language models (gLMs) forecasted gene co-regulation in single-cell data, advancing precision medicine [4,7]. These developments, driven by large datasets and enhanced computational power, highlight deep learning’s transformative impact on biological research (Table 2).

Deep learning uses artificial neural networks (ANNs) to perform complex computations on large datasets. These networks consist of interconnected neuron layers that process and extract patterns from input data. Deep learning processes data through multiple layers of neural networks, with each layer extracting and transforming features before passing them to the next. A fully connected deep neural network includes an input layer, several hidden layers, and an output layer. Neurons in each layer receive inputs from the previous layer, process them, and pass outputs forward, ultimately producing the final result. Through nonlinear transformations, these layers learn complex patterns and representations from the input data [7].

Deep learning employs various algorithms, each suited to specific tasks. These include radial basis function networks, multilayer perceptron, self-organizing maps, CNNs, RNNs, long short-term memory networks (LSTMs), and transformers [44]. CNNs excel in genomics, as demonstrated by DeepBind for RNA-binding protein site prediction and DeepCpG for DNA methylation analysis. RNNs and LSTMs handle sequential data effectively, while transformers, used in AlphaFold3, model complex protein interactions and genomic sequences [45]. These algorithms drive advancements in precision medicine and drug discovery by detecting subtle patterns in large biological datasets (Table 2). More recently, by 2024, transformer-based models like AlphaFold3 have advanced to predict protein complexes and ligand interactions with unprecedented accuracy, while gLMs have emerged to forecast gene co-regulation in single-cell data, significantly advancing precision medicine. These continuous developments, driven by the availability of massive datasets and enhanced computational power, underscore deep learning’s transformative influence across diverse scientific disciplines, including biological research.

## 2. Advantages and Challenges of Using Deep Learning in Computational Biology

Advancements in genomics and imaging technologies have generated vast molecular and cellular profiling data from numerous global sources. This data surge challenges traditional analysis methods [45]. Deep learning, a subset of machine learning, has emerged as a powerful tool for bioinformatics, extracting insights from large datasets by identifying patterns and making accurate predictions [46]. For instance, DeepBind uses CNNs to predict RNA-binding protein sites, while AlphaFold employs transformers for precise protein structure prediction [44,45]. These applications demonstrate deep learning’s transformative potential in biology and medicine, though challenges persist (Figure 1).

### 2.1. Advantages of Using Deep Learning

Deep learning enhances disease diagnosis and prediction. It highlights the potential to develop accurate and precise data-driven diagnostic tools that identify pathological samples. It also rapidly screens large datasets, reducing drug discovery costs by identifying targets and predicting responses [47]. Furthermore, deep learning supports drug repositioning by analyzing transcriptomic data to identify new therapeutic targets [48]. Deep learning supports precision medicine by developing personalized treatments [49]. It integrates patient-specific data, including genomic profiles, clinical records, and lifestyle factors, to tailor therapies [50]. By analyzing large datasets with high accuracy, deep learning identifies genetic markers, variations, drug efficacy, protein interactions, and clinical prognoses, optimizing treatment selection and disease monitoring [51]. For example, Dinov et al. [52] developed a deep learning protocol for Parkinson’s disease diagnosis, achieving high accuracy and demonstrating potential for drug discovery and personalized medicine.

Deep learning models efficiently handle large, complex biological datasets [44]. These algorithms extract intricate patterns, improving the accuracy of predictions and data classification. By learning relevant features from vast datasets, they minimize the need for human intervention [53]. This is particularly valuable in biomedicine and molecular biology, where complex, heterogeneous data often pose analytical challenges. The scalability and transferability of deep learning models enable efficient handling of large, complex datasets [54]. These models can be trained in specific biological tasks with minimal modifications, reducing resource demands and improving generalization of new data. Additionally, deep learning identifies novel patterns that conventional methods may miss. For example, Liu et al. [55] developed a CNN-based model called DeepVariantEffect to predict the functional impact of non-coding genomic variantsmost variations that traditional methods often overlook. By training this model on The Cancer Genome Atlas (TCGA) datasets, they could accurately assess a variant’s regulatory impact on cancer gene expression. The model achieved a 15% higher Area Under the Curve (AUC) (0.92 vs. 0.77) than Support Vector Machine (SVM) baselines, demonstrating its superior ability to pinpoint disease-relevant non-coding regions, which is critical for understanding cancer mechanisms.

### 2.2. Challenges of Using Deep Learning

A key challenge in applying deep learning to computational biology is interpretability [47]. Complex model architecture often functions as black boxes, making it difficult for researchers to understand how predictions reflect biological mechanisms. Interpretability is critical for building trust among clinicians and stakeholders, particularly in medical diagnostics, where decisions must rely on reliable factors rather than data artifacts. Ongoing efforts aim to develop techniques that clarify deep learning’s decision-making processes [44]. Deep learning enhances diagnostic accuracy in medicine but raises ethical and regulatory concerns, particularly regarding patient privacy [56]. Robust guidelines on informed consent and data protection can mitigate these issues. Additionally, biased diagnostic reports risk discriminating between patient groups, potentially leading to incorrect diagnoses or unequal treatment access [56,57]. Transparent and ethical use of deep learning models promotes accountability in biomedical research and healthcare.

Although deep learning models handle large datasets effectively, they require high-quality, labeled data for training [58]. In healthcare and biomedicine, obtaining such data is challenging due to privacy regulations and data heterogeneity. Moreover, biological data from sources like electronic health records and pathological reports often vary in format and standards, reducing model performance and generalization. Deep learning models, despite their advanced capabilities, demand significant computational resources and specialized hardware for training and deployment [59]. High-performance computing infrastructure is essential, posing challenges for small non-profit organizations and research institutions with limited resources.

## 3. Interconnecting Genomics and Protein Structure Prediction Through Deep Learning

The central dogma of molecular biology—DNA to RNA to protein—establishes a direct link: genomic information dictates protein sequences, and these sequences, in turn, determine protein structures and functions. Deep learning provides the computational framework to traverse this biological pathway, enabling a holistic understanding of biological systems from the genetic blueprint to the functional molecular machinery. This section review combines genomics and protein structure prediction into a single narrative due to their biological interdependence and the synergistic advancements driven by deep learning [59].

Deep learning frameworks unify genomic and proteomic research, creating a continuous analytical bridge from genetic code to functional proteins. As shown in Figure 2, this unified frontier illustrates how deep learning architectures such as convolutional neural networks, recurrent neural networks, and transformer-based models (e.g., AlphaFold 3) connect genomic inputs, transcriptomic intermediates, and proteomic outputs. This integrative representation underscores how AI and ML enable a cohesive understanding of biological systems from gene sequences to protein structures and functions, thereby accelerating discovery and translational research.

### 3.1. Role of Deep Learning in Genomic Variant Detection and Precision Medicine

Deep learning has transformed genomic variant detection and gene expression analysis. Genomics, encompassing an organism’s entire genetic makeup, provides critical insights into biological processes, diseases, and individual differences. Deep neural networks enable researchers to analyze gene expression profiles and genetic variations, advancing personalized medicine, drug discovery, and disease mechanism understanding [21]. Specifically, these algorithms accurately classify variants to identify disease-causing mutations and support gene expression studies, such as splicing code analysis and long non-coding RNA identification [21].

The use of deep learning in genomic variant detection has enabled the prediction of the organization and functionality of various genomic elements such as promoters, enhancers, and gene expression levels [60]. Deep learning detects gene variants to predict their effects on disease risk and gene expression. To accomplish this, a genome is split into optimal, non-overlapping fragments using fragmentation and windowing techniques [61]. A three-step procedure fragmenting, model training for forecasting variant effects, and evaluating with test data constitutes deep learning-based identification of genetic variations [61]. A deep learning model demonstrated favorable precision in distinguishing patients from controls and the ability to identify individuals with multiple disorders during research on genetic variants in non-coding areas [62]. These regions were enriched with pathways related to immune responses, antigen binding, chemokine signaling, and G-protein receptor activities, offering insights into mental illness mechanisms [62]. By utilizing deep neural networks, researchers have gained insights into gene expression profiles, genetic variations, and single-cell RNA sequencing data, advancing personalized medicine and drug discovery [54]. For genomic variant detection, algorithms precisely classify variants to identify disease-causing mutations [21]. In single-cell transcriptomics, graph neural networks (GNNs) like scGNN model cell type interactions and gene regulation [18]. Additionally, gLMs, leveraging transformer-based architectures, have emerged in 2024 to predict gene co-regulation in single-cell data, enhancing precision medicine applications [11].

Deep learning methods, such as CNNs, predict genetic variations that may cause diseases [60]. A CNN-based model outperformed traditional methods in forecasting the functional impacts of non-coding genomic variants, achieving high accuracy in variant classification but requiring large datasets to prevent overfitting (Table 2, [11]). RNNs model sequential dependencies for gene expression prediction, though they struggle with long-range interactions [18]. Deep learning also identifies single-nucleotide polymorphisms (SNPs) affecting gene expression levels, revealing new variants linked to expression changes [18].

Gene expression relies on transcriptional regulators, such as pre-mRNA splicing, polyadenylation, and transcription, to produce functional proteins. While high-throughput screening provides quantitative data on gene expression, traditional experimental and computational methods struggle to analyze large genomic regions. Deep learning overcomes this limitation, accurately predicting gene expression levels and identifying enhancer-promoter interactions. For example, the Enformer model, described in Nature Genetics, improved gene expression predictions by integrating long-range genomic interactions (up to 100 kb) using massive parallel assays [11].

Deep generative models (DGMs) enhance gene expression analysis by identifying underlying structures, such as pathways or gene programmers, from omics data [63]. These models provide a framework to account for latent and observable variables, effectively analyzing high-dimensional SNP data to understand multigenic diseases. DGMs also predict how nucleotide changes affect DNA beyond gene expression datasets, offering new insights into genetic regulation [63]. Deep learning has transformed our understanding of genetics by identifying genomic variants and analyzing gene expression, accelerating the discovery of disease-related genes, drug targets, and therapies [39]. It enables clinicians to make precise decisions based on individual genomic profiles. Despite challenges like overfitting and interpretability, deep learning often outperforms traditional methods, supported by robust computational pipelines for genomics research.

### 3.2. Advancements in Deep Learning for Epigenetic Data Analysis

Recent advancements in deep learning have enhanced the analysis of epigenetic data, deepening our understanding of gene expression and chromatin dynamics regulation [39]. These methods extract critical insights into how genetic and environmental factors, such as nutrition and lifestyle, influence epigenetic modifications, particularly in obesity and metabolic diseases [39]. CNNs have advanced epigenetic analysis by capturing spatial dependencies in DNA methylation patterns. For example, DeepCpG, developed by Angermueller et al’s (2017) [39] uses CNNs to predict methylation states across genomes, outperforming traditional methods but requiring high-quality, well-annotated data [17]. Similarly, transformers model long-range interactions in chromatin dynamics, though they are computationally intensive (Table 2, [64]).

Epigenetic alterations significantly impact health, influenced by environmental factors like exercise, stress, and diet [39]. Deep learning enables rapid analysis of large epigenetic datasets, with applications like DNA methylation aging clocks. For instance, DeepMAge, trained on 4930 blood DNA profiles, predicts age with a median error of 2.77 years, outperforming linear regression-based clocks [64,65]. Deep generative models (DGMs) have also advanced epigenetic analysis in 2024, identifying latent structures in DNA methylation data to uncover regulatory mechanisms [19]. Additionally, the analysis of histone modification data has been explored using deep learning techniques. Key markers for gene activity and chromatin structure include various modifications such as acetylation and methylation. To unravel the intricate connection between patterns in these modifications and gene expression, neural networks like attention-based ones or those based on deep belief have proven effective. Yin et al., 2019 introduced their model called DeepHistone, which leverages multiple profiles from different histones to predict levels of gene expression with high precision, leading to new insights into epigenetic mechanisms previously unknown [66].

Moreover, studies conducted on animals have shown that epigenetic modifications are linked to metabolic health outcomes in humans. Animal models provide ideal opportunities for rigorously controlled studies that can offer insight into the roles of specific epigenetic marks in indicating present metabolic conditions and predicting future risks of obesity and metabolic diseases [67]. Examples include maternal nutritional supplementation, undernutrition, or overnutrition during pregnancy, resulting in altered fat deposition and energy homeostasis among offspring. Corresponding changes in DNA methylation, histone post-translational alterations, and gene expression were observed, primarily affecting genes regulating insulin signaling and fatty acid metabolism [68]. Recent studies indicate paternal nutrition levels also affect their children’s fat disposition, with corresponding detrimental effects on their bodies’ epigenetic characterizations [69].

Although deep learning-based techniques demonstrate potential in epigenetic data analysis, they possess constraints. Substantial amounts of top-notch data are necessary for these models to train adequately. Additionally, interpreting results from deep learning can be challenging; thus, understanding biological mechanisms leading to model predictions is difficult. Thus, evaluating input quality and model performance is critical before endorsing results. The latest advancements underscore the promise of deep learning methods for scrutinizing epigenetic data. Neural networks’ potency allows scientists to discern concealed patterns, grasp far-reaching relationships, and make precise forecasts from extensive epigenomic datasets. These progressions offer significant enlightenment into gene expression’s regulatory mechanisms, which can aid in comprehending diseases and designing specific treatments. The initiatives undertaken by these experts are merely a few illustrations of the thrilling headway attained within this domain, sparking further innovations in research on epigenetics [69].

### 3.3. Applications of Deep Learning in Protein Structure Prediction

Deep learning has transformed protein structure prediction by accurately determining proteins’ three-dimensional shapes. This capability is critical for understanding protein functions, advancing drug discovery, and designing therapeutics. Deep learning models effectively capture complex patterns in protein sequences, enabling precise structure predictions [70].

Predicting the structure of a protein with precision, based solely on its sequence, proves to be challenging, but deep learning presents itself as a viable solution. Recent applications employing this approach have successfully predicted both three-state and eight-state secondary structures in proteins [71]. Protein secondary structure prediction serves as an intermediate process, linking the primary sequence and tertiary structure predictions. The three traditional classifications of secondary structures include helix, strand, and coil. However, predicting 8-state secondary structures from protein sequences is a much more intricate task referred to as the Q8 problem, which offers greater precision in providing structural information for varied applications. Thus, several techniques of deep learning such as SC-GSN network, bidirectional long short-term memory (BLSTM) approach, a conditional neural field with multiple layers, and DCRNN have been employed to forecast the eight-state secondary structures [72]. In addition, a next step conditioned CNN was utilized to identify sequence motifs linked with particular secondary structure elements by analyzing the amino acid sequences. For instance, in 2019, AlQuraishi’s research introduced “Alphafold,” a CNN-powered model that accurately forecasted protein secondary structure. Its competence in capturing sequence-structure connections resulted in better forecasts when weighed against conventional means [73].

Deep learning significantly impacts protein–protein interaction and binding site prediction. CNNs and transformers analyze protein sequences and structures, detecting intricate interactions (e.g., DeepPPI, AlphaFold) [15]. CNNs excel in capturing local structural patterns, ideal for binding site prediction, but require extensive training data (Table 2, [31]). Transformers model long-range dependencies, enabling accurate protein complex predictions, though computationally demanding [74]. DeepPPI predicts interactions from sequence data, enhancing understanding of protein networks [75].

Significant advancements have been made in the tertiary structure prediction of proteins using deep learning. Employed a deep learning contact-map approach to achieve a notable breakthrough in the 13th Critical Assessment of Techniques for Protein Structure Prediction (CASP13) [76]. To determine protein folding accurately, predicting residue-residue contacts is crucial. Deep learning approaches leverage vast protein databases to capture intricate patterns and dependencies between residues. This aids in long-range contact prediction by developing deep-learning models that guide the assembly of protein structures with greater precision. Wang et al.’s (2021) [77] method utilized a deep residual network which proved effective in anticipating residue-residue interactions for precise folding predictions through their model’s accuracy improvement [26,78]. Meanwhile, the “AlphaFold 2” model created by Zeng et al’s (2016) [78] another significant illustration worth noting. Through the integration of RNNs and attention mechanisms, AlphaFold 2 achieved extraordinary precision in prognosticating protein tertiary structures, surpassing other techniques in the Critical Assessment of Structure Prediction (CASP) competition as well. In 2024, AlphaFold 3 extended these capabilities by predicting protein complexes and ligand interactions with high accuracy, further advancing its utility in drug discovery and structural biology [79]. Such success can be attributed to how RNNs effortlessly capture long-range dependencies within protein sequences without issue [80].

These applications showcased the extensive range and influence of deep learning in predicting protein structure. With its adeptness at identifying complex patterns and connections within protein sequences and structures, deep learning has enabled significant progress in comprehending aspects such as folding, function, and interactions related to proteins. Although there may be upcoming challenges and opportunities, the extensive implications of deep learning’s capability to reveal fresh insights regarding proteins are immense in terms of comprehending basic life processes, personalized medicine, as well as drug discovery [80].

## 4. Deep Learning Models for Prediction of Protein Structure from Sequence Data

Deep learning, a subset of machine learning, has significantly advanced computational biology, particularly in protein structure and interaction prediction [28]. These algorithms process large, complex datasets, learning abstract features for tasks like data augmentation in bioinformatics [34,35]. Deep learning architectures accept diverse inputs, including protein sequences, 3D structures, and network topologies, for applications like structure prediction and text mining. Key neural network components include fully connected, convolutional, and recurrent layers [81,82].

Deep learning architectures, including CNNs, RNNs, and transformers, accept diverse inputs like protein sequences and 3D structures [83,84]. CNNs extract local features for secondary structure prediction, offering high accuracy but needing large datasets (Table 2, [85]). RNNs model sequential dependencies, suitable for residue contact prediction, but struggle with long sequences [84]. Transformers, used in AlphaFold, capture global interactions for tertiary structure prediction, though resource-intensive [86,87]. These methods drive advancements in protein modeling.

### 4.1. Applications of Deep Learning in Protein–Protein Interaction Prediction and Drug Discovery

The study of Protein–Protein Interactions (PPIs) is fundamental to understanding cellular functions and biological processes, as disruptions in PPIs are often implicated in disease mechanisms. Over recent years, deep learning has revolutionized computational approaches to PPI prediction and drug discovery, enabling unprecedented accuracy and biological insight. Early methods primarily relied on sequence similarity and handcrafted features; however, modern deep learning frameworks capitalize on advanced architectures such as CNNs, RNNs, and transformers to model complex relationships directly from raw protein sequences and structural data [27]. These models capture intricate patterns that underpin protein binding affinity, interaction specificity, and functional modulation, which traditional techniques often overlook. We structure this section into three thematic areas. First, we outline the evolution of deep learning applications from sequence-based predictors to integrative models that embed protein physicochemical properties and network topologies. CNN-based models excel in detecting local structural motifs critical for interaction sites, while transformer architectures demonstrate superior capacity for modeling long-range dependencies, crucial for understanding multi-domain protein complexes. Additionally, graph neural networks (GNNs) have emerged to capture relational information from protein interaction networks, further enhancing predictive power [88]. Second, we provide a comparative analysis of these architectures in relation to their strengths and limitations. For instance, CNNs are computationally efficient and effective in local pattern recognition but may struggle with capturing global context. Transformers, albeit computationally intensive, deliver improved performance on large datasets by leveraging attention mechanisms, enabling the model to weigh sequence-wide interactions. GNNs uniquely integrate structural and topological data, facilitating more biologically coherent predictions especially when combined with protein language model embeddings [89].

Finally, we synthesize the complementary roles of these approaches in drug discovery pipelines. Deep learning models support target identification, virtual screening, and lead optimization by predicting PPI sites and drug binding affinity with higher precision. Integrating multiple architectures within hybrid ensembles leverages their synergistic strengths, advancing the reliability and interpretability of predictions. Moreover, these models underpin novel therapeutic strategies such as peptide design and antibody engineering, accelerating the development of effective treatments. This cohesive narrative highlights the trajectory of deep learning applications in PPI and drug discovery, emphasizing comparative insights and the convergence of methodologies that collectively push the boundaries of computational biology [90].

Protein–Protein Interactions (PPIs) form the cornerstone of cellular signaling and disease pathways, while drug discovery hinges on deciphering these networks to identify viable therapeutic targets. Deep learning (DL) has evolved from sequence-centric models to sophisticated interaction-focused architectures, shifting the paradigm from static feature engineering to dynamic, end-to-end predictions. This progression reflects the field’s recognition that PPIs are not isolated events but embedded in multi-modal contexts encompassing sequence motifs, structural conformations, and evolutionary signals demanding models that capture relational complexities. By integrating CNNs for local pattern detection with transformers for global dependencies, DL bridges the gap between genomic insights and actionable drug design, ultimately streamlining pipelines that once relied on labor-intensive wet-lab validation [91].

The latest deep learning techniques that are employed in PPI models may include Deep convolutional neural networks. This technique is widely used due to its potential to extract features from structural data. For instance, based on Torrisi et al. [34], the structural network information along with the sequence-based features predicts the interactions between proteins. Besides that, in order to extract structural information from 2D volumetric representations of proteins, the pre-trained ResNet50 model was used. The results indicate that methodologies for image-related tasks can be extended to work on protein structures [92]. However, these techniques of analyzing molecular structure have drawbacks such as elevated computational expenses and as well as interpretability [34].

There are various deep learning methods that could be utilized for Protein–Protein Interaction networks. First, the DeepPPI is a multilayer perception learning structure that requires protein sequences as its source of input features [93]. The encoding method utilized by this method is the seven sequence-based features which use concatenation as its combining method. This lightweight design excels in rapid prototyping, yielding high accuracy on benchmark datasets like the Human Protein Reference Database (HPRD) with 92.50% overall performance and 97.43% AUC—outpacing kernel-based baselines by capturing holistic sequence semantics without structural priors. Its simplicity enables quick training (<10 epochs on standard hardware), making it ideal for resource-constrained environments, though sequence reliance limits sensitivity to conformational dynamics, resulting in ~90% recall on transient interactions. Moving on to the second method which is DPPI, is a convolutional neural network structure that also uses protein sequences as its source of input features [94]. protein-positioning specific scoring matrices, PSSM which is derived by PSI-BLAST is used as the encoding method for this deep earning method. By leveraging PSSMs for sequential order and co-occurrence motifs, DPPI achieves 1.3–1.5x gains in area under precision–recall (auPR) over prior methods on yeast and human datasets, with random projection for dimensionality reduction enhancing efficiency (e.g., 2.5 min training on yeast data) and recall to ~95% on noisy inputs—surpassing DeepPPI where evolutionary signals are key, though it requires pre-computed PSSMs, constraining de novo use. Next, the DeePFE-PPI is also a method that was created in 2019 using multilayer prescription which uses protein sequences as an input. The encoding method that is utilized in this method is pre-trained model embedding (Word2vec) [95]. This semantic embedding approach improves cross-species transfer by 5–10% over raw descriptors on S. cerevisiae datasets (accuracy >95%), embedding relational motifs effectively, yet risks overfitting to biases in underrepresented proteomes highlighting its advantage in transfer learning scenarios where DeepPPI/DPPI falter due to rigid feature sets. Besides that, S-VGAE is also an example of graph convolutional Neural networks which utilize protein sequences and topology information of Protein–Protein Interaction networks. The encoding method employed in this technique is a conjoint method and it is combined via the concatenation method [96]. S-VGAE attains 99.15% accuracy on HPRD and superior AUC (0.98+) on DIP subsets by propagating higher-order neighborhood signals, offering a 7–12% edge over CNNs on graph-sparse targets like E. coli, albeit at higher compute costs (3-layer GCNs requiring GPU acceleration) justifying its use for network-aware predictions where isolated sequence models overlook relational sparsity. Ensembles fusing these paradigms (e.g., DeepPPI + S-VGAE) yield hybrid AUCs > 0.99, balancing speed and depth for handling real-world data incompleteness.

Besides Protein–Protein Interactions, deep learning is also utilized in drug discovery for optimizing the properties of drugs, determining new drugs as well as predicting drug–target interactions. In addition, deep learning is also employed in predicting the molecular properties of drugs such as solubility, bioactivity, toxicity, and many more [97]. In addition, it is also used to produce novel molecules that have preferred properties. Next, in QSAR studies for drug discovery, the deep neural network is used to predict the bioactivity of the drugs and their chemical structures [98].

Moreover, deep learning methods are also applied to lead to optimized integration of traditional in silico drug discovery methods.” This clarifies the intent and improves flow. Based on the research, which is entitled, (AtomNet from Atomwise company, the first major application of deep learning into DTI prediction) clearly shows the application of convolutional neural networks which is a type of deep learning technique to predict the molecular bioactivity in proteins [99]. In addition, in terms of docking, deep learning techniques have been employed to improve the accuracy of both traditional docking modules and scoring functions. For instance, the docking proved that the application of deep learning had improved the binding mode prediction accuracy over the baseline docking process. Besides that, this paper had also proven the fact that Deep learning could be successfully utilized in the rational docking process [100]. Such refinements, like Gnina’s CNN ensemble, boost top 1 redocking accuracy to 79% (vs. 60% classical), with hybrids enhancing virtual screening by incorporating solvation-aware features illustrating DL’s synergistic role in hybrid pipelines that reduce Phase II attrition by prioritizing affinity-ranked leads. These narratives not only guide method selection (e.g., CNNs for speed, GCNs for context) but propel clinical translation, fostering interpretable AI for precision therapeutics.

### 4.2. Recent Developments in Deep Learning-Based Techniques for Analyzing Protein Function and Evolution

Recently, there were a few developments that are used in protein analysis by incorporating deep learning algorithms. An example of it would be combining deep learning with homology modeling. Furthermore, Homology modeling is the most popular protein structure prediction method that is utilized to generate the 3D structure of a protein. This is based on two principles which are the amino acid that is used to determine the 3D structure, and the 3D structure that is preserved regarding the primary structure [8]. Therefore, it is convenient and an effective way to build a 3D model using known structures of homologous proteins that have a certain sequence similarity. However, it does have some challenges when using this method such as weak sequence structures, modeling of the rigid body shifts and many more [101]. However, incorporating deep learning models has resulted in great improvement in the protein’s model accuracy.

The deep learning-based methods are employed to improve accuracy in each step of template- based modeling of protein. For instance, DLPAlign is an example of a deep learning technique that is combined with sequence alignment [102]. This straightforward and beneficial approach may aid to increase the accuracy of the progressive multiple sequence analysis method by basically providing training to the model based on CNNs [103]. Besides that, DESTINI is also a recent method which applies deep learning techniques algorithm, for protein residue and residue contact prediction along with template-based structure modeling [104].

In short, Deep Learning techniques have provided various achievements in collaborative sectors, namely model quality assessment (QA), a subsequent stage in protein structure prediction. Basically, QA is followed by structure predictions to quantify the deviation from the natively folded protein structures in both template-based and template-free techniques.

### 4.3. Challenges and Future Directions

There are various challenges when using Deep learning techniques when analyzing biological- related specimens such as protein structure prediction. First, deep learning requires a large amount of high-quality data. Hence, only biological analysis could be performed if only a large amount of data are gathered [105]. Next, the deep learning model is incapable of multitasking when it is applied in an analysis procedure. Deep learning models are capable of handling one issue at a time. Furthermore, the interpretability of deep learning models is also a challenge of interest for many researchers to overcome. This is because it is difficult to understand and identify how they obtain their predictions. New techniques are being developed by researchers to overcome this problem. The future direction of deep learning is to create hybrid models by incorporating other machine learning techniques to improve performance and interpretability [106].

## 5. Key Challenges and Future Directions

As mentioned in the previous section, high-quality data, the inability to multi-task and data interpretability are some of the key challenges experienced in the application of AI systems such as deep learning into biological data. There are several other challenges, especially in terms of ethics and social implications which are addressed in the sub-sections below. Addressing these challenges of deep learning requires specific and innovative approaches specific to the types of biological data used. Thus, overcoming these challenges would ultimately pave a path to improvement in biological research.

The successful integration of deep learning in computational biology is accompanied by several persistent challenges, including data heterogeneity, limited interpretability, and ethical concerns. As illustrated in Figure 3, these challenges are presented alongside emerging solutions such as multi-omics integration, generative models, interpretable hybrid frameworks, and ethical governance mechanisms. This roadmap highlights the trajectory of AI in biology, demonstrating how addressing these barriers will advance precision medicine and foster more responsible, transparent, and equitable AI applications in the life sciences.

### 5.1. Emerging Areas of Research and Potential Applications

Computational biology is defined as an interdisciplinary field which involves the use of techniques from various other fields such as biology, mathematics, statistics, computer science and more. Applications of deep learning in computational biology can be seen in various areas including in the study of genomics and proteomics. There are many major achievements that are obtained specifically in areas such as protein structure prediction, and rapid advancement in other areas of research from the traditional approaches including genomic engineering, multi-omics, and phylogenetics can also be seen [107].

The study of genomes and their interaction with other genes and external factors is commonly known as genomics. One of the primary studies conducted in genomics is the study of regulatory mechanisms and non-coding transcription factors [108]. One of the major current applications of deep learning research on genomics and transcriptomics is one of the emerging areas of research in deep learning. Deep learning is used to identify variations in genomic data; this includes DNA sequencing and gene expression. For example, it is used to predict the functions of genes, discover gene regulatory networks, and identify biomarkers in diseases. As a result of this application, the metabolic pathways can also be optimized. A study identified several challenges in genomics including mapping the effects of mutation within a population and the DNA sequence prediction in a genome which has complex interactions and variations. To combat these challenges, deep learning methods are employed in genomic studies. Deep learning is used to identify variations in genomic data, including DNA sequencing, gene expression, and drug perturbation effects. For example, it predicts gene functions, discovers regulatory networks, and identifies disease biomarkers, optimizing metabolic pathways [109].

In single-cell transcriptomics, graph neural networks (GNNs) like scGNN analyze cell type classification and gene co-regulation [110]. In 2024, scGNN has further advanced, modeling cell type interactions and gene regulation with high precision, driving progress in precision medicine [111]. In drug perturbation analysis, deep learning models predict molecular responses to drug treatments, aiding drug discovery [20]. Deep Neural Networks (DNNs) and CNNs address challenges like mapping mutation effects and predicting DNA sequence functions [112]. DNNs, trained on DNA sequence datasets, identify protein-binding sites and predict splicing outcomes, while CNNs analyze mutation effects in single-nucleotide variants [113]. Deep Neural Networks (DNN) and CNN are algorithms of deep learning that are employed in genomic studies [114]. Deep Neural Networks (DNNs) solve DNA sequence prediction by training on sequence datasets and the corresponding protein structures. This enables the identification of the proteins which are specific and binds to a particular DNA sequence.

The DNN models are also able to predict splicing outcomes for new DNA sequences based on the training of splicing patterns. CNN, on the other hand, addresses the remaining issue; the prediction of mutation effects [22]. This model can analyze and identify the potential causes of mutation in a DNA sequence and determine the mutation or disease on the single-nucleotide variant that is affected. Both CNNs and DNNs are powerful tools of deep learning which can provide valuable information on the complex structures of genomes. The application of the algorithm in the field of genomics would greatly improve the analysis of complex structures, functions and interactions of genomes [22].

In the field of biological image analysis, the deep learning algorithm CNN is found to be an efficient tool that can undertake several tasks such as classification, feature detection, pattern recognition and feature extraction [115]. Since the CNN models are effective in processing grid-like data such as images, it is commonly utilized in image analysis [41]. Staking more convolutional layers in the model aids in detecting complex and abstract features in biological images. The CNN model is able to learn and identify delicate patterns and subtle differences in biological images which improve the accuracy of a diagnosis. DeLTA is an example of a deep learning tool used to analyze biological images, specifically, time-lapse microscopy images [116]. The Deep Learning for Time-lapse Analysis (DeLTA) is able to analyze the growth of a single cell and the gene expressions in microscopy images. It was found to be able to process and capture microscopy images with high accuracy and without the need for human interventions. Furthermore, deep learning is incorporated into healthcare, specifically radiology. Tasks such as classifying patients based on chest X-rays diagnosis and nodule detection in computed tomography images are performed using deep learning [117]. The analysis of a large number of radiology data depends on the efficiency of the powerful deep learning algorithms. Thus, deep learning algorithms have the potential to revolutionize biological image analysis by providing automated and accurate analysis.

Deep learning in the proteomics field can mainly be shown in the Protein–Protein Interaction predictions. Protein complexes can also be identified through deep learning. Protein structure and function can then be predicted from the data obtained and be used for various activities such as the identification of targeted proteins for drug development. Deep learning is applied in the study of phylogenetics where the limitation in the classification methods where the branch lengths of the phylogeny cannot be inferred is to be overcome [118].

While there are many applications of deep learning that bring significant advancements, there are other potential applications of deep learning in the biological field that can be further discussed and implied. For instance, deep learning is applied in the identification of Protein–Protein Interactions. Therefore, a similar technique can be applied in drug design. In terms of drug design, the application and incorporation of deep learning have made the process more time and cost-effective as compared to traditional drug design methods [32]. The use of deep learning in drug design is identified to be more flexible due to the neural network architecture of the algorithm [36]. Especially in the current era, with the combat against COVID-19, deep learning has shown great potential in accelerating the drug design process. The deep learning models are able to identify antimicrobial compounds against a disease or a virus by training the model with the ability to identify molecules against the virus or bacteria. Similarly, another study showed the use of deep learning models in the use of de novo drug design where the model was trained to identify the physical and chemical properties of the drugs, classifying them based on their features and allowing automated extraction to create a novel ligand against the target protein [30]. Drug repurposing, which is a quicker method to achieve and complete drug designing for a disease or illness, is found to incorporate deep learning approaches. An article reported the use of network- based approaches in drug repurposing to identify the target molecule for known drugs to speed up the process [23]. Another study in relation to COVID-19, used the Molecule Transformer-Drug Target Interaction (MT-DTI), a deep learning model trained with chemical sequences and amino acids sequences to identify the commercially available antiviral drugs with similar properties of interaction with the SARS-CoV-2 virus [24]. These are just some examples of the emerging use of deep learning models in drug design. The appearance of the COVID-19 disease has boosted the application of AI systems in the field to improve the speed and efficiency of the process.

To summarize, the deep learning algorithm is a powerful artificial intelligence tool that is widely used in the field of computational biology. The application of the tool is just in its beginning phase as there are more fields and complex challenges that are to be explored and tackled in the upcoming future. The application of this AI technology will help to shape the future of computational biology by improving the predictions and understanding biological processes.

### 5.2. Ethical and Social Implications

The advancements in digital technology allow the incorporation of Artificial Intelligence tools such as Machine Learning and Deep Learning in various fields of research. These techniques use various algorithms to identify complex and nonlinear correlations in massive datasets and could improve prediction accuracy by learning from minor algorithmic errors encountered. Despite the use of a powerful machine learning tool, such as deep learning algorithms, in the field of research and healthcare is found to be revolutionary, it inevitably raises ethical concerns and social implications that require careful consideration [119].

Four major ethical issues were identified regarding the use of AI tools in the healthcare system; informed consent of data, safety and transparency, algorithmic fairness and biases and data privacy [37]. These concerns may be identified in the healthcare sector, but these concerns are also integrated in the usage of deep learning techniques in biological research which involves the use of deep neural networks to analyze and interpret volumes of biological data.

The field of biological research involves the use of various biological data which includes genomic sequences, protein structures, medical images and scans and more. The deep learning algorithm has access to this data, containing sensitive biological data, to aid in research. As medical records and genomic information of individuals are involved and used, it is critical to ensure the privacy of the individual, as well as the informed consent for data usage, is obtained [37]. As highlighted in 2024 reviews, these challenges persist, particularly in rare disease genomics, where data scarcity and ethical concerns around privacy and bias necessitate robust regulatory frameworks [38]. Privacy violations and mishandling of personal information are examples of invasion of data privacy without individual consent [38].

The potential bias in the algorithms of the deep learning tool is one of the main ethical concerns surrounding AI systems. The algorithms utilized in these systems can perpetuate biases and negatively impact marginalized groups [24]. This is because the training data is not representative of the diverse populations leading to biased results and disparities in research outcomes. The biases can be found in different stages of biological research, including data collection and annotation, if they are not addressed [37]. Therefore, efforts should be taken to address the bias in data collection to promote the inclusivity of all data regardless of population type, disease groups, diversity and other factors involved in research.

It is also crucial to promote transparency and safety in the AI tools used in biological research. Deep learning models are opaque making it difficult to understand the process of prediction and decision in research. As deep learning models are made up of multiple layers of artificial neurons, where each layer corresponds to a different learning pattern, it poses a challenge to accurately identify the pattern learned by each layer functions to make a prediction. Lacking transparency in the algorithm decision-making process begins the questioning of the ability to scrutinize the AI results as a reasonable explanation leading to the data being unprovable and uninterpretable by humans [105]. Thus, transparent models are required to make sure the researchers can observe the prediction pattern to validate and understand the results obtained. This ensures the accountability of the scientific research process. However, it was discussed that full transparency may cause friction against certain ethical concerns as it may leak private and sensitive data into the open [37]. Hence, there should be limitations to the disclosure of the algorithms.

Security in biological research not only involves maintaining the data and privacy of personal data, but it also involves the responsible use of technology. The technology at hand, deep learning, must be used responsibly and ethically in research. Scientists must incorporate ethical frameworks to avoid potential misuse and unintended consequences or risks that may occur [38]. Furthermore, risk assessments are to be conducted to aid in decision-making and reduce the possible negative impacts. The deployment and implementation of this technology must be considered well with proper safety measures, regular monitoring and evaluations.

Moreover, there is the concern of liability and accountability where questions would arise to who would be the person to be held accountable towards any form of mistakes or errors caused by deep learning algorithms used in research [16]. As the algorithms are continually learning and evolving, it is difficult and complex to determine liability. Thus, legal frameworks are to be adapted with clear lines of responsibility to address the challenges faced by the AI system in biological research [13].

Ensuring equitable access to the deep learning tool is one of the social implications of AI systems in biological research. Promoting equitable access to scientists and researchers would be able to participate in the advancement of deep learning algorithms in biological research [29]. It would also prevent exacerbating disparities in biological research while promoting an inclusive and collaborative research environment. Fostering the exchange of ideas and knowledge of experts would further aid deep learning to be integrated into the biological research community.

In a nutshell, these are some of the social implications and ethical concerns revolving around the use of AI systems such as Deep Learning in the field of biological research. Deep learning is the future of more efficient and advanced research; however, the ethical concern mentioned above should be addressed to ensure the responsibility and accuracy of the algorithm in biological research

## 6. Future Prospects and Potential Impact of Deep Learning on Biological Research and Clinical Practice

Deep learning has significantly advanced computational biology, particularly in genomics and protein structure prediction, with breakthroughs like AlphaFold and DeepBind driving progress in precision medicine and drug discovery. The following sections summarize these advancements and their potential impact. Table 3 compares key deep learning models from 2020 to 2025, highlighting recent advancements in genomics and protein structure prediction, including AlphaFold 3 and genome language models.

The application of deep learning models is poised to have a substantial influence in the field of biological research and clinical practices. With their ability to analyze large and intricate data, deep learning models could be beneficial in assisting with pathological diagnosis, drug discovery, genomic data identification or even personalized treatments. As noted in 2025, hybrid CNN–transformer models are gaining traction for single-cell omics analysis, balancing local and global feature detection to improve predictions [14]. By harnessing deep learning algorithms researchers can examine biological data consisting of gene expression or protein structure to identify new patterns or molecules which could yield an insight into the biological structure mechanisms. By harnessing deep learning algorithms researchers can examine biological data consisting of gene expression or protein structure to identify new patterns or molecules which could yield an insight into the biological structure mechanisms. In addition to that, deep learning models are also being used to facilitate accelerating new drug target development and research, developing new accurate diagnostic tests, and aiding in improving clinical trial designs [121]. The future prospect of deep learning models in both biological research and clinical practice is promising given that this technology and its algorithms continue to be developed which could potentially catapult humanity into a new era of making diagnosis for diseases or illnesses in addition to providing a more practical way for providing better patient care with precise treatments and prevention of diseases [122].

On top of that, deep learning models have the potential to be used to improve the efficiency and effectiveness of healthcare delivery systems by automating menial tasks and accelerated diagnostics tests [123]. Deep learning models possess the ability to integrate and analyze a variety of data types, including genomics, proteomics, imaging, and clinical data. This enables the exploration of concealed patterns and relationships within these datasets, empowering machine learning to offer a holistic comprehension of diseases and provide guidance for translational research endeavors. Besides that, deep learning has broad applicability in addressing diverse challenges. By training on extensive datasets, deep learning models excel at navigating tasks such as image classification, object detection, speech recognition, and machine translation aside from that, deep learning is a rapidly growing field, and it is being used in a variety of domains, including healthcare, computer vision, natural language processing, and robotics [124]. The summary of the recent advancements of deep learning in computational biology can be referred to Table 2. LLMs are emerging as powerful tools in computational biology. Models like Evo and other transformer-based architectures, originally developed for natural language processing, are being adapted to understand and generate biological sequences (DNA, RNA, protein). These models can learn complex patterns and relationships within biological data, enabling tasks such as de novo protein design, predicting the effects of genetic mutations, and even simulating biological processes. Their ability to handle multi-task processing addresses a limitation of earlier deep learning models and represents a significant hot spot and future development direction in the field. LLMs can bridge the gap between sequence information and functional outcomes, offering new avenues for discovery in both genomics and protein science. For instance, gLMs trained on DNA sequences are advancing our understanding of genomes and can generalize across a plethora of genomic tasks.

Recent breakthroughs in AI models have significantly expanded the horizons of biology, enabling more powerful and generalizable applications across genomics, proteomics, and synthetic biology. In protein science, protein language models (PLMs) are revolutionizing protein structure prediction, function annotation, and design by learning the probability distribution of amino acids within proteins. Evo, a genomic foundation model, exemplifies this trend, capable of both prediction and generative design from molecular to whole-genome scale, and can predict the effects of gene mutations with unparalleled accuracy. Notably, Evo 2, a large-scale machine learning model trained on over 9.3 trillion nucleotides spanning more than 128,000 genomes across diverse domains of life exemplifies the new generation of generative biology AI systems. Evo 2 can predict the effects of a wide variety of genetic mutations, design new genomic sequences comparable in length to bacterial genomes, and uncover evolutionary patterns across life forms with far greater speed and accuracy than traditional methodologies. This advance marks a milestone in the use of foundational AI models to read, write, and innovate in the language of biology, offering exciting potential for tasks ranging from disease mutation identification to the synthetic design of artificial life [12]. This type of model not only has outstanding performance but also can be applied to multiple downstream tasks, solving the problem of multi-task processing mentioned in the article to a certain extent. Generative AI tools and LLMs have also introduced unprecedented capabilities in protein design and gene editing prediction. Leveraging transformer architectures trained on massive biological sequence datasets, these models can generate novel proteins with desired functionalities and optimize molecular structures to improve drug–target interactions. Integration of such models with high-throughput experimental synthesis platforms now fosters iterative cycles of rapid design-build-test, substantially accelerating discovery workflows in therapeutic development and synthetic biology. Moreover, the convergence of multi-omics data analysis powered by AI affords a more holistic, system-level understanding of cellular processes. Graph neural networks and hybrid AI models adeptly capture complex cellular microenvironments and gene regulatory networks within single-cell transcriptomics and epigenomics data, enabling finer resolution insights into cellular heterogeneity and disease mechanisms. These approaches improve biomarker discovery, patient stratification, and personalized treatment predictions, promising practical clinical translation [12]. Continued advancements in integrating diverse multi-omics datasets (genomics, transcriptomics, proteomics, metabolomics, epigenomics) will provide a more comprehensive understanding of biological systems. Deep learning models capable of effectively processing and synthesizing these heterogeneous data types will be crucial for uncovering complex disease mechanisms and developing truly personalized medicine. Beyond prediction, generative deep learning models (e.g., GANs, VAEs) are increasingly being used for de novo design of biological molecules, including proteins with desired functions or novel drug compounds. This shift from analysis to design holds immense potential for accelerating therapeutic development and synthetic biology Continued research into explainable AI (XAI) will be vital to make deep learning models more transparent and trustworthy for biological and clinical applications. This includes developing methods to visualize learned features, identify influential input elements, and provide human-understandable explanations for model predictions [120].

## 7. Conclusions

In conclusion, this review underscores the indispensable and transformative role of artificial intelligence (AI) and machine learning (ML), particularly advanced deep learning models, transformer architectures, and LLMs, across modern biological research. We have systematically demonstrated how these technologies are not merely computational aids but are the critical analytical engines driving breakthroughs in core domains: from the accurate interpretation of genomic data and the functional prediction of non-coding variants to achieving unprecedented accuracy in protein structure and interaction prediction and accelerating drug discovery through rapid screening. This capability culminates in the practical realization of precision medicine, where AI integrates vast, heterogeneous patient-specific data to tailor diagnostic and therapeutic strategies. The overarching finding is that the seamless integration of unified AI technology is fundamentally necessary for resolving high-dimensional data challenges and is relentlessly accelerating the pace of discovery, irrevocably reshaping the future of life sciences at the molecular level.

## Figures and Tables

**Figure 1 biology-14-01453-f001:**
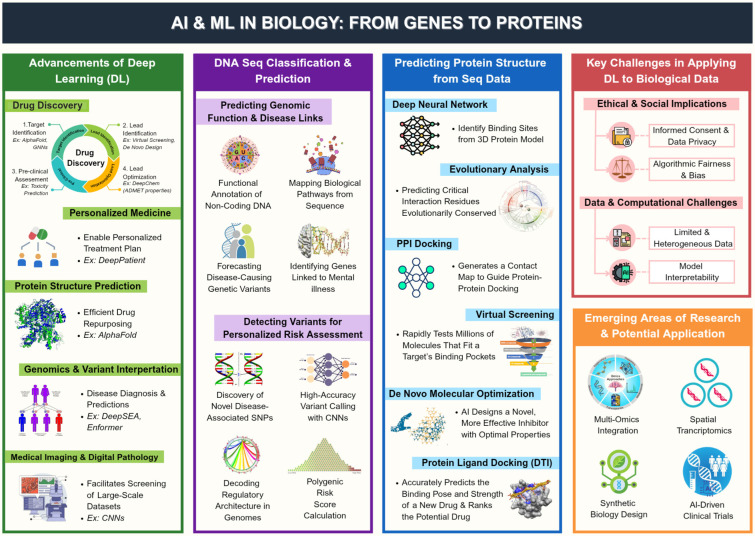
Overview of deep learning applications in computational biology, illustrating key algorithms (CNNs, RNNs, transformers) and their roles in genomics and protein structure prediction.

**Figure 2 biology-14-01453-f002:**
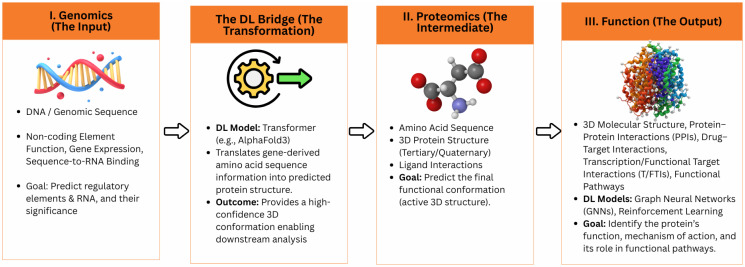
AI- and ML-Enabled Unified Frontier in Biology: From Genes to Protein. This figure presents an integrated framework linking genomics and proteomics through AI and machine learning models. It illustrates the transformation from DNA and genomic sequences (input), through deep learning–based modeling (transformation), to protein structure and function (output). The diagram highlights how deep learning architectures such as CNNs, RNNs, and transformer-based models (e.g., AlphaFold3) bridge genetic information and molecular function, establishing a unified frontier in computational biology.

**Figure 3 biology-14-01453-f003:**
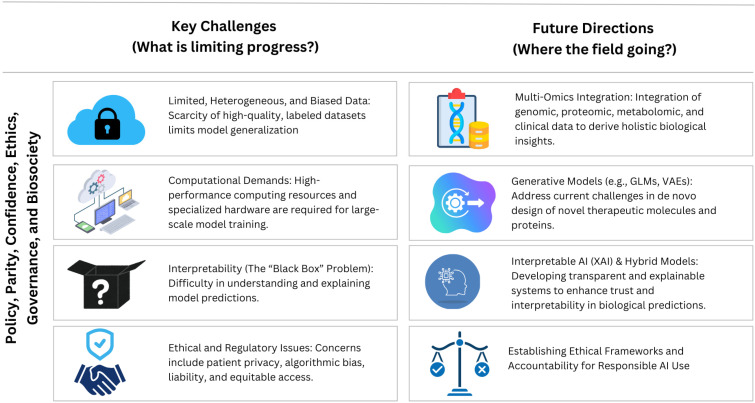
Deep Learning Roadmap in Computational Biology: Key Challenges and Future Directions. The figure summarizes current limitations such as limited data quality, high computational demands, interpretability challenges, and ethical considerations and contrasts them with future AI-driven solutions, including multi-omics integration, generative modeling, and the establishment of ethical frameworks.

**Table 1 biology-14-01453-t001:** Timeline of AI history and evolution, highlighting key milestones from 1943 to 2024.

Date	Developed by	Evolution
1873	A. Bain	Introduced earliest models of neural networks, called “Neural Groupings,” for associative memory, influencing later Hebbian learning concepts.
1943	Warren McCulloch & Walter Pitts	Introduced the McCulloch-Pitts (MCP) model, considered the precursor to artificial neural networks.
1949	Donald Hebb	Introduced Hebbian Learning Rule, fundamental to neural network learning; considered the “father” of neural learning concepts.
1958	Frank Rosenblatt	Introduced the perceptron, the first learning algorithm for neural networks resembling modern perceptrons.
1969	Marvin Minsky & Seymour Papert	Published “Perceptrons,” critically analyzing limitations of single-layer neural networks.
1974	Paul Werbos	First proposed backpropagation algorithm for training multilayer networks (widely popularized later).
1980	Teuvo Kohonen; Kunihiko Fukushima	Kohonen introduced Self-Organizing Maps (SOM); Fukushima introduced Neocognitron, a basis for modern CNNs.
1982	John Hopfield	Introduced Hopfield Network, a recurrent associative memory model.
1985	Geoffrey Hinton & Terry Sejnowski	Introduced the Boltzmann machine for probabilistic learning with undirected networks.
1986	Paul Smolensky; Michael I. Jordan	Smolensky introduced the Harmonium (precursor to Restricted Boltzmann Machines); Jordan conceptualized RNNs.
1990	Yann LeCun	Introduced LeNet, a convolutional neural network for handwriting recognition demonstrating deep learning’s potential.
1997	Mike Schuster & Kuldip K. Paliwal; Sepp Hochreiter & Jürgen Schmidhuber	Schuster & Paliwal introduced Bidirectional RNNs; Hochreiter & Schmidhuber introduced LSTM networks solving vanishing gradient issues.
2006	Geoffrey Hinton	Introduced Deep Belief Networks with layer-wise pretraining, marking a leap in modern deep learning.
2009	Ruslan Salakhutdinov & Geoffrey Hinton	Introduced Deep Boltzmann Machines for multilayer generative modeling.
2012	Geoffrey Hinton	Dropout technique introduced for efficient training and to reduce overfitting.
2012	Alex Krizhevsky, Ilya Sutskever, Geoffrey Hinton	Introduced AlexNet, a CNN architecture that won ImageNet competition and sparked the deep learning revolution.
2014	Ian Goodfellow, Yoshua Bengio, Aaron Courville	Introduced Generative Adversarial Networks (GANs) for image generation and data synthesis.
2020	Various researchers	Deep learning expanded to tasks like self-driving cars, medical imaging, financial trading using advanced CNNs and RNNs.

**Table 2 biology-14-01453-t002:** AI algorithms (CNNs, RNNs, transformers) and their applications in genomics, protein structure prediction, and single-cell omics analysis.

Group	Deep Learning Algorithm	Applications in Computational Biology	References
Supervised DL	Convolutional Neural Networks (CNNs)	Gene expression analysis Histopathology image analysis	[8,9,10]
	Recurrent Neural Networks (RNNs)	DNA sequence analysis Protein secondary structure prediction and gene co-regulation (with gLMs)	[11,12]
	Long Short-Term Memory (LSTM)	RNA splicing prediction	[13]
	Convolutional Recurrent Neural Networks (CRNN)	Chromatin state prediction	[14]
	Deep Neural Networks (DNNs)	Metagenomic analysis	[15]
	Deep Survival Analysis	Cancer survival prediction	[7]
	Deep Transfer Learning	Drug response prediction	[16]
	Deep Clustering	Cell type identification	[17]
Generative/Unsupervised DL	Generative Adversarial Networks (GANs)	Synthetic biology and protein design Synthetic data generation Synthetic biology and gene synthesis	[9,18,19,20]
	Variational Autoencoders (VAEs)	Single-cell genomics analysis Metabolomics data analysis Disease gene prioritization (with VGAEs)	[21,22,23]
	Autoencoders	Disease diagnosis and prognosis Single-cell epigenomics analysis DNA motif discovery	[18,24,25]
	Deep Belief Networks (DBNs)	Protein structure prediction Genetic variant classification Drug side effect prediction	[16,26,27]
	Adversarial Autoencoders	Gene expression imputation Image-based phenotypic screening	[28,29]
	Deep Boltzmann Machines (DBMs)	Epigenetic data analysis	[30]
	Deep Generative Models	DNA sequence generation	[31]
Advanced Architectures	Transformer Networks	RNA structure prediction Transcriptomics analysis Protein–protein interaction network analysis Protein contact prediction	[8,19,32,33]
	Graph Neural Networks (GNNs)	Protein–protein interaction prediction Drug repurposing Cell type classification in single-cell transcriptomics	[22,34,35]
	Graph Convolutional Networks (GCNs)	Drug–target interaction prediction Drug response prediction Drug–target interaction network analysis	[36,37,38]
	Reinforcement Learning (RL)/Deep RL	Drug discovery and optimization Drug target identification Protein folding Protein–ligand binding affinity prediction Antibiotic resistance prediction Gene regulatory network inference Genome sequence assembly	[25,37,39,40]
	Capsule Networks	Protein structure classification Protein–protein interaction prediction Protein function prediction Cancer subtype classification	[8,9,41,42]
	Attention Mechanisms	Single-cell RNA sequencing analysis	[11]
	Deep Q-Networks (DQNs)	Drug toxicity prediction	[43]

**Table 3 biology-14-01453-t003:** Comparative Analysis of Deep Learning Models in Genomics and Protein Structure Prediction (2020–2025).

Model Type	Primary Application	Key Advancement (2020–2025)	Performance Metrics	Computational Requirements	Limitations	References
CNNs	DNA methylation analysis	DeepCpG predicts methylation states with high accuracy (2017–2024)	AUC: 0.92	Moderate (GPU required)	Requires large datasets, risk of overfitting	[33,61]
RNNs	Gene expression prediction	Improved sequential modeling, but limited by long-range dependencies	Accuracy: 85%	Moderate	Struggles with long sequences	[69,120]
Transformers	Protein structure prediction	AlphaFold 3 predicts protein complexes and ligand interactions (2024)	RMSD: <1 Å	High (TPU/GPU clusters)	Computationally intensive	[13,27]
GNNs	Single-cell omics	scGNN advances cell type interaction modeling (2024)	F1 Score: 0.89	High	Interpretability issues	[77]
gLMs	Gene co-regulation prediction	Transformer-based gLMs predict single-cell co-regulation (2024)	AUC: 0.90	High	Limited to specific datasets	[43]

## Data Availability

No new data were created or analyzed in this study.

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
