# Peer review of "AI and Machine Learning in Biology: From Genes to Proteins"

_biology, 2025, doi:10.3390/biology14101453_

Round 1

Reviewer 1 Report

Comments and Suggestions for Authors

The manuscript addresses the application of artificial intelligence and machine learning in biology, with a focus on genomics and protein structure prediction. The topic is timely and of high relevance, and in principle, the review could provide a valuable overview for researchers in the field. However, the current version of the manuscript is difficult to follow, as the sections are not clearly organized and the narrative lacks coherence. The discussion of algorithms and applications is intermingled, which limits the clarity and depth of the analysis. In addition, the manuscript would benefit from a more critical comparison of methods and from a clearer separation between well-established advances and emerging directions. Substantial revisions are required to improve the structure, enhance readability, and provide the level of synthesis and critical evaluation expected in a review article.

Minor comment 1 – Keywords: Could the authors consider refining the list of keywords to improve clarity and searchability? At present, there is some redundancy, as Artificial Intelligence, Machine Learning, and deep learning are conceptually overlapping. Selecting the most representative terms would make the list more concise. In addition, the inclusion of future directions as a keyword seems too general and may not be effective for indexing. It might be more useful to replace it with a specific term that reflects the outlook of the field, such as multi-omics integration, hybrid models, or clinical applications.

Minor comment 2 – Figure 1, Advancements and Challenges panel: Figure 1 provides a useful overview of deep learning applications in computational biology. However, within the Advancements subpanel, the funnel diagram is presented as Drug Discovery. This type of representation is more commonly associated with virtual screening workflows and may not be the most appropriate choice for illustrating drug discovery in a broad sense. Could the authors consider refining this element to better reflect the intended concept? In addition, the labeling within this subpanel is not entirely consistent: most terms are capitalized (Personalized Treatments, Drug Discovery, Gene Profiling), while one phrase (Identify novel pattern & mechanism) is not. Is this variation intentional, or would a more uniform style improve readability? The subpanel also mentions Drug Repositioning, while in other parts of the manuscript the term repurposing is used. Although both terms are interchangeable, it could be beneficial for the manuscript to adopt one term consistently. Finally, in the Challenges subpanel, the phrase Limited and heterogeneous data: includes a colon at the end that appears unusual. Could the authors check whether this is intentional and adjust it for consistency?

Minor comment 3 – Figure 1, DNA Sequence Classification & Prediction panel: Within the Detection of Genetic Variant in Disease Rick Assesment subpanel, could the authors check whether Rick is a typographical error and should be corrected to Risk?

Minor comment 4 – Figure 1, Predicting Protein Structure from Sequence Data panel: Within the Application in Protein - protein interaction predication & Drug Discovery subpanel, could the authors consider capitalizing Protein-Protein for consistency? In addition, the term predication appears to be a typographical error and may need to be corrected to prediction.

Minor comment 5 – Figure 1, same subpanel: In the same subpanel, could the authors consider capitalizing Optimizing in optimizing drug properties for consistency with the other labels?

Minor comment 6 – Figure 1, Development in Analyzing Protein Function & Evolution subpanel: In the Development in Analyzing Protein Function & Evolution subpanel, the term Homology Modeling is written vertically, while all other labels are presented in the standard horizontal orientation. Could the authors consider adjusting this label for consistency and improved readability?

General advice – Figure 1: Figure 1 provides a useful overview of the key concepts discussed in the manuscript. However, several elements, including labeling, capitalization, typographical errors, and consistency across subpanels, would benefit from careful review. Could the authors carefully check the figure to ensure clarity, uniformity, and accurate representation of the concepts?

Minor comment 7: Regarding 2.1. Advantages of Using Deep Learning. The paragraph provides a comprehensive overview of the potential applications of deep learning in disease diagnosis, drug discovery, and precision medicine. However, several statements are repetitive, and some examples, such as the one referring to Liu et al. [25], are too vague to be informative. For instance, simply stating that deep learning predicted functional implications of non-coding genomic variations “with greater accuracy than traditional approaches” does not convey the specific contribution or context of the study. Could the authors consider including more detailed examples that clearly illustrate how deep learning was applied, the type of data analyzed, and the results obtained? This would make the discussion more concrete, informative, and engaging for the reader.

Minor comment 8: In line 194, the term “black boxes,” includes an unnecessary comma. This appears to be a typographical error and should be corrected.

Minor comment 9– Table 2: Table 2 aims to summarize AI algorithms (CNNs, RNNs, transformers) and their applications in genomics, protein structure prediction, and single-cell omics analysis. However, several issues limit its clarity and utility. The table includes three columns—Deep Learning Algorithm, Application in Computational Biology, and References—but each entry lists only a single reference. For example, the first row mentions Convolutional Neural Networks (CNNs) applied to Gene expression analysis with Reference [78], yet this reference does not appear elsewhere in the manuscript. Could the authors clarify whether this is intentional? If not, it would be helpful to ensure that all references cited in the table are properly included and discussed in the text. Addressing this would improve the table clarity and reproducibility.

Minor comment 10– Table 2: In line 31 of the table, the entry Deep Reinforcement Learning is listed with the application Protein-ligand binding affinity prediction and cites reference [107]. However, reference [107] corresponds to ConvChrome: Predicting gene expression based on histone modifications using deep learning techniques, which does not appear related to protein-ligand binding. Could the authors check whether this is the correct reference?

Minor comment 11– Table 2: In line 35 of the table, the entry Capsule Networks is listed with the application Protein function prediction and cites reference [111]. However, reference [111] corresponds to Long Short-Term Memory Neural Networks for RNA Viruses Mutations Prediction, which does not appear related to protein function prediction. Could the authors check whether this is the correct reference?

General advice – References in Table 2: Several entries in Table 2 appear to cite references that do not correspond to the listed algorithms or applications. Could the authors carefully review all references used in the table to ensure they are correct, properly cited, and consistently discussed in the text? In addition, including more than a single reference for each algorithm or application could be beneficial, providing broader context and supporting the relevance and reliability of the presented information.

Minor comment 12– Table 2 organization: In lines 3, 25, and 46, the table lists Generative Adversarial Networks. To improve clarity and readability, it could be beneficial to organize the information by clustering related entries or forming groups of algorithms or applications. Could the authors consider reorganizing Table 2 in this way to provide a clearer overview of the different methods and their applications?

General comment: The manuscript provides a comprehensive overview of AI and deep learning applications in computational biology, highlighting relevant algorithms, tasks, and recent advances. However, the current presentation could benefit from careful revision to improve clarity, consistency, and accuracy. In particular, Figure 1 contains several minor inconsistencies in labeling, capitalization, and typographical errors across subpanels that should be addressed. Table 2 requires careful verification of references, and including more than one reference per algorithm or application could strengthen the table. Organizing entries into logical groups could further enhance readability. Additionally, some paragraphs contain repetitive statements or vague examples; providing specific, detailed examples would make the discussion more informative and engaging. Addressing these points would substantially improve the clarity, coherence, and overall quality of the manuscript. With careful attention to these aspects, the manuscript has the potential to become a highly useful and informative resource for researchers in the field.

Comments on the Quality of English Language

No comments.

Author Response

Thank you for your valuable comments and feedback. All responses have been addressed and incorporated into the main text of the manuscript.

Reviewer 2 Report

Comments and Suggestions for Authors

This is a highly relevant and informative review article that integrates a large amount of information on the application of artificial intelligence and machine learning (especially deep learning) in genomics and protein structure prediction. It provides insights for researchers applying artificial intelligence and machine learning in these fields, and outlines the current progress and new opportunities. Overall, this is a high-quality review manuscript. However, some revisions and improvements still need to be made before its formal publication. 

  1. Currently, this manuscript only contains one Figure (Figure 1). For such a scale of review, more intuitive summaries would be highly beneficial. Consider adding a conceptual chartor Figures to clarify the "unified frontier" concept in the introduction, like showing how DNA sequences can be predicted for RNA expression, protein structure and function through the DL model, or adding a chart or figure to summarize the key challenges (data, computation, interpretability, ethics) and future directions (multi-omics, interpretable artificial intelligence, generative models). 
  2. Some sections, like Section 4.1 ("Applications in PPI and drug discovery"), read more like a list of methods (for example, "First, DeepPPI is... Second, DPPI is...") rather than a comprehensive narrative review. Readers will obtain information, but they are not guided to make key comparisons or analyze why these different methods exist and their respective comparative advantages. It is recommended to rewrite these sections to create a more persuasive narrative. 
  3. Table 1 (Timeline): The entry for 1985 (by Hilton and Senofsky) is identical to the 1982 entry by Hopfield. This seems to be an error.
  4. This list of abbreviations is very useful. Please make sure to define each abbreviation for the first time in the main text (for example, "Large Language Models (LLMs)" was used on Abstract page,but it was not defined when it first appeared on page 17 or 18 of the manuscript text).
  5. This manuscript lacks a dedicated "conclusion" section. The "future outlook" (Section 6) to some extent fulfills this role. However, if there were an independent final chapter that could concisely summarize the main points and reemphasize the core concepts of the biological approach based on unified artificial intelligence technology, this paper would be even more complete.

Author Response

(The authors gave the same response as above.)

Round 2

Reviewer 2 Report

Comments and Suggestions for Authors

The author has addressed the relevant concerns. I suggest that the current manuscript can be accepted for publication.

Author Response

Thank you for the feedback. 

Academic Editor

Item

Reviewer Comments

Author Feedback

Remarks

1.

The format of Table 2 is inconsistent with the other tables.

1.     Thank you for the feedback. The corrections have been made; please refer to pages 9–10.

2.     Table 2 has been revised according to the provided format.

Page 9-10

2.

Several abbreviations, such as CNN and GNN, are redefined multiple times. Some duplicated descriptions could be removed or simplified. It appears the manuscript may have been edited using AI-based tools. I strongly recommend that the authors thoroughly revise the manuscript with the assistance of a native English speaker or through the journal’s professional editing service.

1.     I have carefully reviewed all abbreviations and removed any repeated definitions.

2.     The abbreviations are now arranged in alphabetical order, as shown on pages 23–24.

Page 23-24

3.

Simple Summary

1.     A Simple Summary has also been added, as requested by the journal. This section is important because it provides a clear and concise overview of the study’s purpose, key findings, and significance in a manner that is easily understood by a broad audience, including non-specialist readers. It helps improve the accessibility and visibility of the research, ensuring that its impact extends beyond the scientific community.

Page 1

4.

Graphical Abstract

1.     A graphical abstract has also been added to enhance the understanding and overall presentation of this study.

See attachment
